# Long-Term Outcomes of Breast Cancer Patients Who Underwent Selective Neck Dissection for Metachronous Isolated Supraclavicular Nodal Metastasis

**DOI:** 10.3390/cancers14010164

**Published:** 2021-12-29

**Authors:** Shin-Cheh Chen, Shih-Che Shen, Chi-Chang Yu, Ting-Shuo Huang, Yung-Feng Lo, Hsien-Kun Chang, Yung-Chang Lin, Wen-Ling Kuo, Hsiu-Pei Tsai, Hsu-Huan Chou, Li-Yu Lee, Yi-Ting Huang

**Affiliations:** 1Department of General Surgery, Division of Breast Surgery, Chang Gung Memorial Hospital, Chang Gung University Medical College, Linkou Branch, Taoyuan 333, Taiwan; wolf5052@gmail.com (S.-C.S.); kenneth0609@cgmh.org.tw (C.-C.Y.); loyf@cgmh.org.tw (Y.-F.L.); sylvie5285@gmail.com (W.-L.K.); peipeitsai@gmail.com (H.-P.T.); b9002009@cgmh.org.tw (H.-H.C.); 2Department of General Surgery, Chang Gung Memorial Hospital, Keelung Branch, Keelung 204, Taiwan; huangts@cgmh.org.tw; 3Department of Medicine, Division of Medical Oncology, Chang Gung Memorial Hospital, Chang Gung University Medical College, Linkou Branch, Taoyuan 333, Taiwan; CHK0329@seed.net.tw (H.-K.C.); yclinof@cgmh.org.tw (Y.-C.L.); 4Department of Pathology, Chang Gung Memorial Hospital, Chang Gung University Medical College, Taoyuan 333, Taiwan; r22068@cgmh.org.tw; 5Department of Radiation Oncology, Chang Gung Memorial Hospital, Chang Gung University Medical College, Linkou Branch, Taoyuan 333, Taiwan; hyt3784@cgmh.org.tw

**Keywords:** breast cancer, supraclavicular nodal metastasis, selective neck dissection, survival

## Abstract

**Simple Summary:**

Very limited studies so far have analyzed the long-term oncologic outcomes of breast cancer patients that developed metachronous isolated supraclavicular nodal metastasis (miSLNM) with no available treatment strategy for the critical issue. The study enrolled 139 miSLNM patients; 61 patients underwent selective neck dissection. In median follow-up of 73.1 months, significantly better 5-year overall survival rate was found in the neck dissection group compared to the no-dissection group (68.9% vs. 57.7%, respectively; HR, 1.77 (1.22–2.55), *p* = 0.003). The findings suggest surgery for miSLNM should be integrated into multimodal therapy of miSLNM, and the restaging of miSLNM as rN3c rather than M1 disease if detected earlier.

**Abstract:**

We retrospectively enrolled 139 patients who developed metachronous isolated supraclavicular lymph node metastasis (miSLNM) from 8129 consecutive patients who underwent primary surgery between 1990 and 2008 at a single medical center. The median age was 47 years. The median follow-up time from date of primary tumor surgery was 73.1 months, and the median time to the date of neck relapse was 43.9 months in this study. Sixty-one (43.9%) patients underwent selective neck dissection (SND). The 5-year distant metastasis-free survival (DMFS), post-recurrence survival, and overall survival (OS) rates in the SND group were 31.1%, 40.3%, and 68.9%, respectively, whereas those of the no-SND group were 9.7%, 32.9%, and 57.7%, respectively (*p* = 0.001). No SND and time interval from primary tumor surgery to neck relapse ≤24 months were the only significant risk factors in the multivariate analysis of DMFS (hazard ratio (HR), 1.77; 95% confidence interval (CI), 1.23–2.56; *p* = 0.002 and HR, 1.76, 95% CI, 1.23–2.52; *p* = 0.002, respectively) and OS (HR, 1.77; 95% CI, 1.22–2.55; *p* = 0.003 and HR, 3.54, 95% CI, 2.44–5.16; *p* < 0.0001, respectively). Multimodal therapy, including neck dissection, significantly improved the DMFS and OS of miSLNM. Survival improvement after miSLNM control by intensive surgical treatment suggests that miSLNM is not distant metastasis.

## 1. Introduction

Supraclavicular lymph node metastasis (SLNM) of breast cancer is a clinical challenge for most patients presenting with de novo M1 disease, and it has poor outcomes, although its incidence rate was 3.7% to 8% [1,2,3,4], more prevalent in patients with high disease burden, such as more than four positive axillary nodes, and those with axillary level II or III nodal involvement. A recent clinicopathological study demonstrated the association of lymphovascular invasion with regional lymph node metastasis and systemic metastasis, suggesting that the anastomotic pathway of systemic metastasis from primary breast cancer was through regional lymph nodes [5]. Another single hospital database analysis revealed that regional nodal involvement usually precedes systemic metastatic dissemination [6]. Both studies suggested that regional nodal metastasis (including SLNM) should be categorized as locoregional disease rather than systemic disease, and that survival improvement can be achieved by intensive local control. Obviously, improvement of the survival of patients with SLNM should depend on early detection with intensive clinical surveillance and the aggressive multimodal approach of systemic therapy and radiotherapy [7]. However, a high risk of residual disease in the supraclavicular fossa has been reported even after aggressive radiotherapy [7]. Moreover, there are limited studies on surgical extirpation, instead of radiotherapy, incorporated in the multimodal therapy of SLNM [8,9,10,11].

SLNM can be classified as synchronous and metachronous based on the timing of the recurrence, and the latter is defined as ipsilateral neck nodal relapse occurring 6 months after primary tumor surgery. Synchronous SLNM (sSLNM), with the TNM staging of N3_C_ category in the updated American Joint Commission on Cancer (AJCC) staging system, has been discussed in most studies. Metachronous isolated SLNM (miSLNM) was defined as rN3c if metastatic lymph nodes are in the supraclavicular fossa, a triangle defined by the omohyoid muscle and tendon, the internal jugular vein, and the clavicle and subclavian vein, and lymph nodes outside this triangle are considered to be non-lower cervical nodes staging as M1 [12]. To the best of our knowledge, only a few papers have discussed metachronous isolated SLNM (miSLNM) [13,14,15,16,17,18]. Herein, we report a retrospective review of miSLNM with long-term follow-up to evaluate the role of surgery as a multimodal therapy to improve survival outcomes based on our previous research papers on SLNM [19,20].

## 2. Materials and Methods

### 2.1. Study Population and Data Collection

We conducted a retrospective analysis of 8129 consecutive patients who underwent surgical treatment for primary breast cancer at Chang Gung Memorial Hospital, Linkou, Taiwan between 1990 and 2008. All patients received regular in-hospital follow-up care, including routine physical examinations and laboratory testing every 3 to 6 months in the initial 2 years, every 6 months in the next 3 years, and then annually. Chest radiography, abdominal ultrasonography, breast sonography, and mammography were conducted annually and repeated if necessary. Selected patients also underwent bone scans and whole-body computed tomography if the suspicious disease progressed clinically. Tumor stages were defined according to the 8th edition of the American Joint Committee on Cancer criteria.

Criteria for inclusion included women over 20 years of age, and miSLNM was confirmed by cytology or pathology after core needle biopsy. Patients were followed up until November 2020. Patients with concurrent distant metastases upon initial diagnosis of SLNM were excluded. Data were collected for the analysis of demographic characteristics, surgical pathology, recurrence patterns, treatment strategies, and survival outcomes. Adjuvant chemotherapy, hormonal therapy, and radiotherapy were administered to patients following the institutional treatment guidelines, as previously described [21]. Among the patients with recurrence, locoregional recurrence was defined as recurrence in the chest wall or regional lymph nodes without distant metastasis. miSLNM was defined as recurrence in the ipsilateral supraclavicular lymph nodes with or without recurrence in the chest wall or regional nodes and no evidence of distant metastasis after a 6-month follow-up period. Distant metastasis was defined as any distant organ metastasis identified by diagnostic imaging or histopathology. Distant metastasis-free survival (DMFS) was defined as the time from the diagnosis of miSLNM to the date of the identification of distant metastasis, and post-recurrence survival (PRS) was defined as the date from relapse in the neck to death or last follow-up. Overall survival (OS) was defined as the time from the date of primary tumor surgery to death or last follow-up. This study was approved by the ethics committee of the Chang Gung Medical Foundation.

### 2.2. Treatment after miSLNM

Patients who underwent selective neck dissection (SND) in addition to chemotherapy and/or radiotherapy were assigned to the SND group, whereas those who received only chemotherapy and/or radiotherapy were assigned to the no SND group. The decision of whether or not patients should undergo SND was dependent on the clinician’s judgment and preference, and SND was performed via a low transverse skin incision over the neck. The platysma muscle was separated anterior to the sternocleidomastoid muscle. With careful identification and preservation of the vascular structures and nerves, all suspicious lymph nodes and fibroadipose tissues were removed. The dissection completely removed the neck level IV lymph nodes around the internal jugular vein extending superiorly from the greater auricular nerve and inferiorly to the clavicle, and level III and V lymph node dissection was performed according to preoperative imaging and intraoperative findings. Radiotherapy to the neck in the SND group was administered if metastatic lymph nodes could not be completely removed during surgery; otherwise, radiotherapy was only administered to the chest wall and internal mammary chains for those who developed chest wall or regional lymph node metastasis.

### 2.3. Statistical Analysis

Patient characteristics at initial primary tumor surgery and neck nodal metastases were summarized as N (%) for the categorical variables and median (interquartile range, IQR) for the continuous variables by treatment group. Chi-square test or Fisher’s exact test was used to compare the categorical variables, and Wilcoxon rank-sum test or t-tests were used to compare the continuous variables, as appropriate. Kaplan–Meier curves were used to visualize unadjusted DMFS, PRS, and OS, and the log-rank test was used to test for differences among groups. Cox proportional hazards regression analysis was used to estimate the association between treatment groups and OS after adjustment for covariates. We report hazard ratios (HRs), 95% confidence intervals (CIs), and *p*-value significant at 0.05. All analyses were conducted using SPSS software (IBM Corp. Released 2011. IBM SPSS Statistics for Windows, version 20.0. Armonk, NY, USA: IBM Corp.).

## 3. Results

There were 139 women with miSLNM in the cohort of 8129 breast cancer patients between 1990 and 2008, including 119 patients with miSLNM alone, 11 with local relapse, and 9 with axillary nodal relapse. A total of 122 patients with miSLNM concurrent with distant metastasis at diagnosis were excluded. Retrospective chart reviews revealed that miSLNM recurrence was detected by self-examination in 6 patients, routine physical examinations in 71 patients, ultrasonography in 44 patients, and computed tomography in 18 patients. The median age was 47 years (IQR 15) and the median follow-up time from the primary tumor surgery was 73.1 months (IQR 87.5). The time from the date of neck relapse to the last follow-up date was 43.9 months (IQR 59.7).

There were 61 patients who underwent SND, and 78 patients did not. The initial clinical characteristics of the primary tumors and treatments are summarized in Table 1. For the patients who underwent SND, the median age was 47 (IQR 15) years, the mean number of dissected nodes was 11.9 ± 10.4 (mean ± sd) and of positive nodes, 7.4 ± 7.8 (mean ± sd), 15 patients were initial node negative, most patients received chemotherapy (85.2%), and only 9 (11.5%) received adjuvant radiotherapy. Twenty-three of the 78 (29.5%) patients with no dissection had received prior radiotherapy.

There were no significant differences in clinical features between the SND group and the no SND group, including the time interval from the primary surgery to neck relapse, largest neck node size, and salvage treatments after neck relapse (Table 2). In total, 72 patients did not receive radiotherapy, and 30 patients in the SND group received radiotherapy after neck nodal relapse.

At 6 years of median follow up, SND improved DMFS, PRS, and OS (Figure 1A–C). The 5-year DMFS, PRS, and OS of the SND group were 31.1%, 40.3%, and 68.9%, respectively, and those of the no-SND group were 9.7%, 32.9%, and 57.7%, respectively (*p* = 0.001). A time interval from primary tumor surgery to neck relapse of less than 24 months was associated with worse DMFS, PRS, and OS (Figure 2A–C). There were no significant differences in DMFS, PRS, and OS between the patients who received both systemic therapy and radiotherapy and those who received systemic therapy alone (Appendix A). Those with concurrent miSLNM and distant metastasis (*n* = 122) had worse PRS in comparison to both the SND group and the no SND group, as well as a worse OS than the SND group (Appendix A).

There were significant differences in the median time interval from neck relapse to distant metastasis between the SND group and the no SND group, 23.7 months (IQR 25.3) and 14.7 months (IQR 20.4) (*p* = 0.009), respectively. The median time intervals from neck relapse to death were 44.7 months (IQR 43.0) and 30.1 months (IQR 46.9) (*p* = 0.047), respectively.

In total, 51 (83.6%) patients developed distant metastasis in the SND group with systemic therapy, with or without radiotherapy, and 48 (78.7%) patients died of the disease (Appendix A). In the no SND group, 75 of 78 (96.2%) patients developed distant metastasis and 73 (93.6%) patients died of the disease.

The improvement in DMFS, PRS, and OS by SND remained significant in the multivariable proportional hazards model (Table 3, Appendix A). Only two variables, no SND group and time interval between primary tumor surgery and neck relapse less than 24 months, had significantly worse DMFS (HR, 1.77; 95% CI, 1.23–2.56; *p*
= 0.002 and HR, 1.76, 95% CI, 1.23–2.52; *p* = 0.002, respectively) and significantly worse OS (HR, 1.77; 95% CI, 1.22–2.55; *p* = 0.003 and HR, 3.54, 95% CI, 2.44–5.16; *p* < 0.0001, respectively) in the multivariate analysis.

## 4. Discussion

Our data demonstrated that in multimodal therapy, SND in addition to systemic therapy was associated with overall survival improvement when compared with no surgery, and the results are consistent with recent studies showing that long-term survival outcomes of patients with sSLNM were improved by multimodal therapy with curative intent [22,23]. As intensive clinical surveillance with modern imaging modalities, such as sonography, to detect subclinical distant nodal relapse earlier and multimodal therapy, including SND and systemic therapy, were associated with improved 5-year DMFS, RFS, and OS, we can treat miSLNM as locoregional relapse rather than systemic metastasis.

Regional nodal recurrences, including those of the axillary, supraclavicular fossa, and internal mammary nodes, are rare events and have been associated with favorable outcomes in recent publications [24,25]. sSLNM was downstaged to cN3c in the AJCC staging system 2002, based on a retrospective study of survival improvement after multimodal therapy in a single institution [26]. After that, the 5-year OS was reported to be approximately 40–45% in most studies of sSLNM [3,10], confirming sSLNM as a form of locoregional disease rather than distant metastasis. However, there is no standard of what kind of treatment should be included in multimodal therapy, and surgical treatment of SLNM was not included in the guidelines from the National Comprehensive Cancer Network (NCCN) [27]. In other studies, surgery of sSLNM was associated with improved survival [9,10], and radical irradiation improved locoregional control but not survival [11].

On the contrary, miSLNM and contralateral axillary lymph node metastasis (CAM) are staged as rN3c or M1 disease (in our series, most of the locations of metastatic nodes were beyond the triangle of supraclavicular fossa and involved the level III and IV and even level V of neck), while many studies have demonstrated that both miSLNM and CAM are associated with a better prognosis than distant metastasis if managed with intensive multimodal therapy with curative intent [18,20,22,25]. Surgical removal of miSLNM is usually neglected in multimodal therapy because it is categorized as systemic disease with inconsistent survival benefit of surgical treatment. A recent cohort study from the Surveillance, Epidemiology, and End Results registries database found that patients with distant lymph node metastasis (DLNM) (including to the neck, contralateral axillary, and internal mammary nodes) had 3-year OS rates that were similar to those of patients with sSLNM (62.67% and 53.46%, respectively), but the OS rates were significantly better than those for patients with distant metastasis (38.21%) [22]. They recommended aggressive locoregional therapies for the disease. Reddy et al. reported a large cohort of miSLNM patients with a 5-year OS of 46%; they also demonstrated that aggressive local therapy with curative intent might improve long-term outcomes [28]. While Pergolizzi et al. observed a 5-year OS of 35% in a prospective multicenter study examining the role of combining chemotherapy and radiotherapy in the treatment of miSLNM [14]. Altogether, miSLNM does not always lead to dismal outcomes, and such patients should be considered good candidates for aggressive therapy with curative intent rather than palliative treatment.

In our long-term follow-up study of miSLNM, the SND group had a significantly longer time interval from neck metastasis to occurrence of distant metastasis compared to the no SND group (23.7 months and 14.5 months, respectively), and the interval from neck relapse to death was also longer in the SND group (44.7 months and 30.1 months, respectively). The survival of our miSLNM patients was comparable to that of patients with sSLNM in the literature [1,9,29], and there were significant differences in the 5-year DMFS between the SND group and the no SND group (31.1% and 9.7%, respectively) and in the 5-year OS, too (68.9% and 57.7%, respectively). Intensive locoregional therapy (radiotherapy or surgery) seems to be mandatory in multimodal therapy to achieve good local control and prevent subsequent distant metastasis [17,18,23]. The curative intent of SND as described in our method should be reinforced by surgical removal of all suspicious nodes in level IV and part of those in level V, not just excision of the gross tumor as was done in most of the other studies [11]. The major surgical complications of SND included nerve or vessel injury, deep vein thrombosis, shoulder function impairment, and rarely, chylous leak. Most of the complications are manageable and preventable with careful preoperative assessment, meticulous surgical technique, and postoperative care.

Radiotherapy has historically been suggested as the only intensive locoregional therapy for SLNM with controversial survival benefits [4,14,26]. Our data demonstrated that only SND and miSLNM that occurred later (>24 months) were significantly associated with better outcomes, while radiotherapy in addition to chemotherapy did not improve DMFS and OS. The randomized phase 3 trial, EORTC 22922/10925, showed no disease-free or overall survival improvement, but a significant reduction in breast cancer mortality and any breast cancer recurrence by elective internal mammary and medial supraclavicular (IM-MS) irradiation [30]. However, the reduction in the incidence rate of SLNM was only 0.9% (1.6% and 2.5% in patients with and those without IM-MS irradiation, respectively).

Chemotherapy for isolated locoregional recurrence of breast cancer was proven to improve survival in the randomized CALOR trial [31]. However, one study demonstrated that newly acquired mutations not seen in the primary tumor with key driver mutation in the synchronous lymph node metastasis result in more resistance to chemotherapy [32]; additionally, 45.5% of SLNM patients still had residual metastatic disease after intensive chemotherapy, and there was no reliable factor to predict pathologically complete response to omit SND [23]. These findings suggest that chemotherapy alone is not adequate to improve survival in locoregional recurrence.

Another factor contributing to better outcomes is early detection of isolated regional nodal metastasis by intensive clinical and ultrasonography surveillance during the follow-up period. The meta-analysis indicated that early detection of breast cancer recurrence during follow-up offers significantly prolonged survival compared to late detection of recurrence [33]. Moon et al. reported that ultrasonographic surveillance of the chest wall and neck for regional nodal relapse after breast cancer surgery had a sensitivity and specificity of 76.9% and 98.7%, respectively [34]. In our institute, ultrasonography was included in the surveillance modalities for breast cancer patients, and the axillary fossa, internal mammary, and supraclavicular fossa were included in the areas to be scanned. In this study, 31.7% of miSLNMs were detected by ultrasonography during routine follow-up.

Basic studies of lymphangiogenesis demonstrated the clinical relationship between lymphovascular invasion and regional lymph node metastasis, and sentinel node biopsy has successfully replaced axillary dissection in the modern era, suggesting that in cancer metastasis, there is orderly spread of the metastatic cells through the regional lymph nodes to the rest of the body’s systems [5,35,36]. Earlier detection of isolated regional nodal disease and curative surgical eradication prevents local event progression to advanced disease and reduces distant metastasis, stressing a plausible future revision of miSLNM, as sSLNM, to rN3c in the TNM staging system.

Disease-free interval of neck relapse >24 months was a significantly better prognostic factor for DMFS and OS in our study, which is consistent with other studies indicating a more indolent disease and favorable tumor biology [37]. However, it was not a confounding factor in the SND and no SND groups in our study, because there was no significant difference in the time interval from primary tumor surgery to neck relapse between the two groups (Table 2).

This study had several limitations. First, it was a retrospective study with a small sample size, and this may have affected the statistical power of our findings. However, randomized trials are difficult to perform owing to the rare incidence rate of miSLNM. Our study is the longest comparative study to investigate the efficacy of SND. Second, the selection bias when choosing patients on whom to perform SND by metastatic nodal burden and the surgeon’s preference may have affected the clinical outcomes. Third, the patients were enrolled from 1990, and there were some missing data and no updated treatments provided, such as anti-HER2 therapy. Fourth, there was an imbalance in the number of patients who received radiotherapy after primary tumor surgery between the two groups because the indication of adjuvant radiotherapy in our hospital treatment guidelines before 2010 was limited to tumor size larger than 5 cm or more than three involved nodes.

## 5. Conclusions

In this cohort study, we found comparable DMFS and OS in patients with miSLNM and sSLNM as well as significantly better survival than in patients with distant metastasis if the patients were treated with multimodal therapy that included curative intent SND.

## Figures and Tables

**Figure 1 cancers-14-00164-f001:**
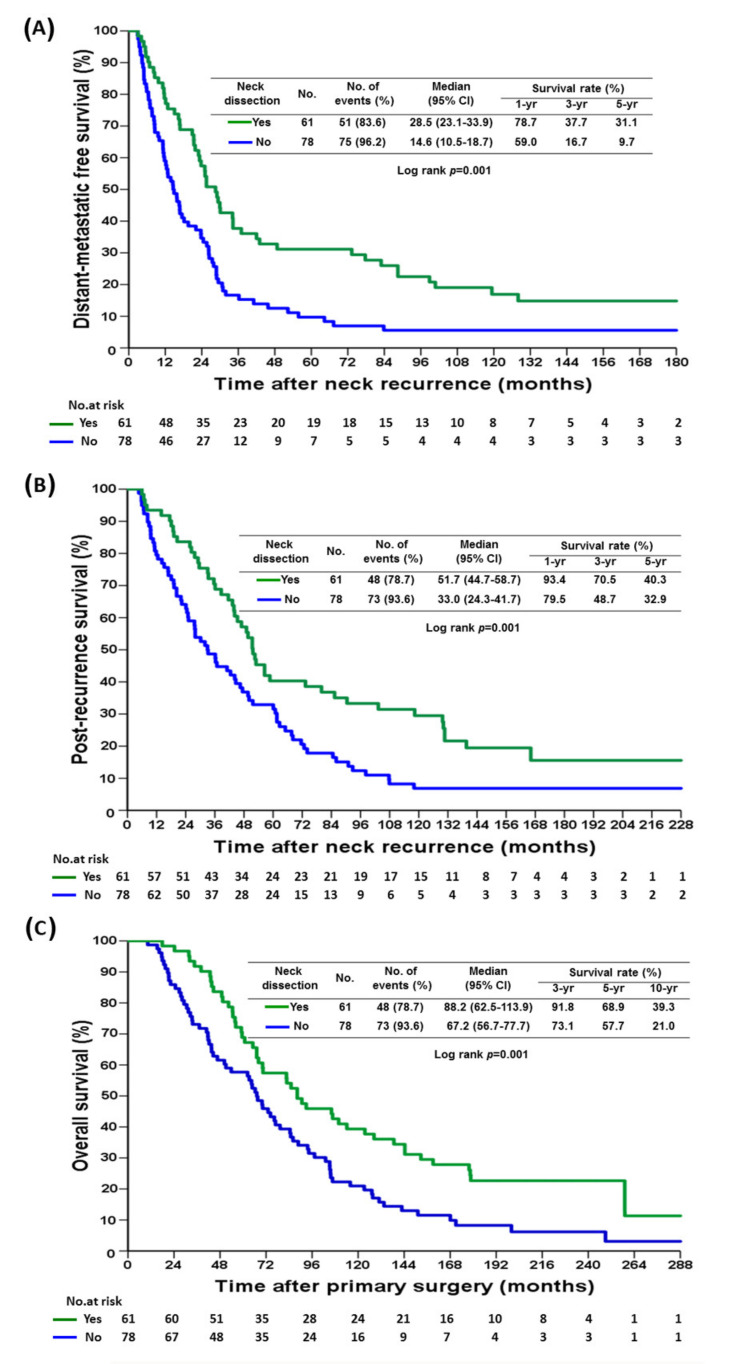
Kaplan–Meier curve of (**A**), distant-metastatic free survival (**B**), post-recurrence survival (**C**), overall survival for miSLNM patients by neck dissection.

**Figure 2 cancers-14-00164-f002:**
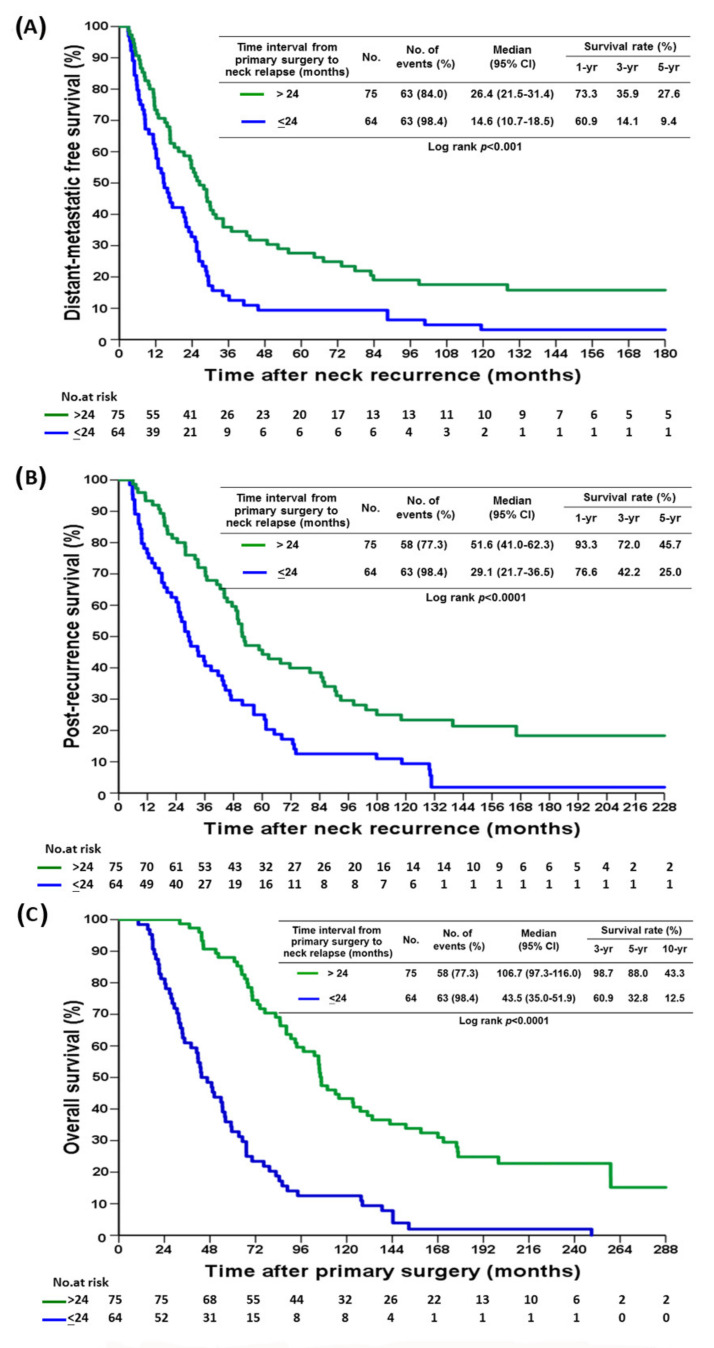
Kaplan–Meier curve of (**A**), distant-metastatic free survival (**B**), post-recurrence survival (**C**), overall survival for miSLNM patients by time interval from primary surgery to neck dissection.

**Table 1 cancers-14-00164-t001:** Initial Clinical Features of Patients with Metachronous Isolated Supraclavicular Lymph Node Metastasis, by Selective Neck Dissection or Not.

Variables		Neck Dissection (n = 61)	No Dissection (n = 78)	*p* Value
Age, years, median (IQR)		47 (15)	45 (16)	0.516
Tumor size, cm, median (IQR)		2.5 (1.1)	3.0 (2.0)	0.053
Operation procedure	Total mastectomy	57 (93.4)	67 (85.9)	0.155
	Partial mastectomy	4 (6.6)	11 (14.1)	
Axillary involvement	Yes	46 (75.4)	59 (75.6)	0.975
	No	15 (24.6)	19 (24.4)	
Estrogen receptor status	Positive	33 (54.1)	42 (53.8)	0.266
	Negative	27 (44.3)	30 (38.5)	
	Unknown	1 (1.6)	6 (7.7)	
Progesterone receptor status	Positive	30 (49.2)	39 (50.0)	0.270
	Negative	30 (49.2)	33 (42.3)	
	Unknown	1 (1.6)	6 (7.6)	
HER-2/neu	Positive	17 (27.9)	21 (26.9)	0.749
	Negative	19 (31.1)	29 (37.2)	
	Unknown	25 (41.0)	28 (35.9)	
SBR grade	1	8 (13.1)	7 (8.9)	0.623
	2	17 (27.9)	29 (37.2)	
	3	23 (37.7)	25 (32.1)	
	Unknown	13 (21.3)	17 (21.8)	
Initial axillary level II dissection	Yes	52 (85.2)	66 (84.6)	0.918
	No	9 (14.8)	12 (15.4)	
Chemotherapy	Yes	54 (88.5)	70 (89.7)	0.818
	No	7 (11.5)	8 (10.3)	
Radiotherapy	Yes	9 (14.8)	23 (29.5)	0.041
	No	52 (85.2)	55 (70.5)	
Hormonal therapy	Yes	32 (52.5)	41 (52.6)	>0.999
	No	29 (47.5)	37 (47.4)	

Figures are numbers with percentages in parentheses, unless otherwise stated.

**Table 2 cancers-14-00164-t002:** Treatment Patterns After Metachronous Isolated Supraclavicular Lymph Node Metastasis, by Selective Neck Dissection or Not.

Variables		Neck Dissection (n = 61)	No Dissection (n = 78)	*p* Value
Time interval from primary surgery, months, median (IQR)		28.7 (35.5)	24.0 (27.0)	0.131
Time interval from primary surgery				
	≤24 months	25 (41.0)	39 (50.0)	0.309
	>24 months	36 (59.0)	39 (50.0)	
Age at relapse, years, median (IQR)		52 (13)	49 (17)	0.233
Largest size of neck node, cm, median (IQR)		1.3 (2.0)	1.3 (1.0)	0.843
Treatment after relapse				
Systemic therapy + radiotherapy		30 (49.2)	26 (33.3)	0.021
Systemic therapy		30 (49.2)	42 (53.8)	
None		1 (1.6)	10 (12.8)	

Figures are numbers with percentages in parentheses, unless otherwise stated.

**Table 3 cancers-14-00164-t003:** Univariate and Multivariate Analysis of Distant-Metastatic Free Survival after Metachronous Isolated Supraclavicular Lymph Node Metastasis.

Variables		No.	Median Survival Time (Months)	95% CI ^*^of Median	*p* ∆Value	HR ^#^	95% CI of HR	*p* Value
Initial clinical features								
Age (years)	≤40	42	22.1	16.6–27.5	0.589	–		
	>40	97	18.1	11.7–24.6				
Tumor size (cm)	≤3	92	22.9	17.7–28.1	0.544	–		
	>3	47	16.8	8.1–25.5				
Axillary involvement	Yes	105	20.8	14.9–26.8	0.180	–		
	No	34	23.7	9.8–37.6				
Estrogen receptor status	Positive	75	17.5	9.7–25.3	0.617	–		
	Negative	57	23.7	16.9–30.5				
Progesterone receptor status	Positive	69	21.8	12.5–31.1	0.692	–		
	Negative	63	21.4	13.6–29.1				
HER-2/neu	Positive	38	25.6	24.1–27.2	0.823	–		
	Negative	48	20.8	13.5–28.2				
SBR grade	1	15	41.0	19.8–62.2	0.087	–		
	2	46	18.1	8.6–27.7				
	3	48	16.8	9.6–23.9				
Axillary level II dissection	Yes	118	22.9	16.8–29.0	0.608	–		
	No	21	14.9	9.1–20.6				
Adjuvant therapy before relapse								
Chemotherapy	Yes	124	21.8	15.9–27.8	0.886	–		
	No	15	14.9	4.4–25.3				
Hormonal therapy	Yes	73	23.3	15.5–31.1	0.247	–		
	No	66	20.8	14.4–27.3				
Radiotherapy	Yes	32	23.7	14.4–33.0	0.650	–		
	No	107	20.8	14.5–27.2				
Clinical features after relapse								
Age at relapse	≤50	70	20.8	13.5–28.2	0.475	–		
	>50	69	22.9	14.0–31.9				
Clinical neck node size (cm)	≤1.3	47	20.8	11.4–30.3	0.403	–		
	>1.3	46	21.3	13.4–29.3				
Selective neck dissection	Yes	61	28.5	23.1–33.9	0.001	1		
	No	78	14.6	10.5–18.7		1.77	1.23–2.56	0.002
Time interval from primary tumor surgery to neck relapse (months)	≤24	64	14.6	10.7–18.5	<0.001	1.76	1.23–2.52	0.002
	>24	75	26.4	21.5–31.4		1		
Chemotherapy	Yes	113	22.9	17.4–28.4	0.614	–		
	No	26	14.5	9.7–19.3				
Hormonal therapy **	Yes	78	24.5	19.5–29.5	0.058	–		
	No	12	9.6	4.5–14.8				
Radiotherapy	Yes	57	26.3	19.5–33.1	0.103	–		
	No	82	16.8	11.2–22.4				

* 95% CI: 95% confidence interval; ∆log rank test; ^#^ HR: hazard ratio; ** Select primary tumor or neck tumor with ER and/or PR (+) cases.

## Data Availability

The data presented in this study are available on request from the corresponding author. The data are not publicly available due to it is collected from our hospital database under ethical regulations.

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
