# Peer review of "Long-Term Outcomes of Breast Cancer Patients Who Underwent Selective Neck Dissection for Metachronous Isolated Supraclavicular Nodal Metastasis"

_cancers, 2021, doi:10.3390/cancers14010164_

Round 1

Reviewer 1 Report

Chen et al. did a comprehensive analysis of survival outcomes for patients with differing interventions suffering from SLNM. The conclusions were interesting and useful for the clinic.

A few minor concerns:

  1. The background could be expanded. What is the occurrence of SLNM? How commonly do breast cancer patients develop SLNM? Is this disease more prevalent in certain populations? The discussion briefly talked about a current clinical trial, but are there any more ongoing for SLNM?
  2. In the discussion, please address any advances in the medical field since 2008 that may have limitations on the data collected and this study. Have treatments/survival changed in the past 12 years?
  3. Line 61, different reference style?
  4. There is something wrong with the two sentences between Lines 68 and 72. The first sentence appears incomplete.
  5. The second line (line 151-152) of the title for Table 1 is confusing. Perhaps there is an unnecessary punctuation here?
  6. Ethical Review should be its own paragraph and not an add-on to ‘Statistical Analysis’. Since an IRB statement is included below the ‘Conclusion’, the added line in the ‘Statistical Analysis’ can be removed.
  7. I would’ve liked to see authors mentioning a reason behind using the cutoff of 24 months for classifying patients based on the time interval from the primary surgery to the neck relapse. I assume that the decision was based on the data that, for non-SND group the median time interval was 24 months. Would the statistical values be notably different if the cutoff was 36 months or 18 months instead of 24 months?
  8. Did the authors try to identify scientific reasoning behind clinicians’ preference for/against SND? Was there any other pathologies involved that prevented SND and could’ve played a role in the distant metastasis?

Reviewer 2 Report

The study discussed the long-term survival of selective neck dissection in patients who developed miSLNM. This study is novel and sound for the future treatment strategy of miSLNM.

Author Response

Thank you for your comment. Your opinion is greatly appreciated.

Reviewer 3 Report

It’s a great manuscript. However, the authors should address the below concerns:

  1. The authors use the selective neck dissection of level III-V. Is it an evidence based management? Any references could support such therapeutic surgery?
  2. The authors mention that “The decision of whether or not patients should undergo SND was dependent on the clinician’s judgement and preference” Was there any patient having miSLNM but without performance of SND? Further, was any reason why the surgeon didn’t perform SND for these patients? The authors should detailed explain it.
  3. In discussion, the authors mention “Surgical removal of miSLNM is usually neglected in multimodal therapy because it is categorised as systemic disease and also the different subspecilization in surgery, as breast surgeons are not obviously trained in head and neck surgery”. The authors should mention if there was any complication of the patient undergoing neck dissection. The authors should not only mention the benefit of SND, but the authors should also mention the disadvantage of SND.

Round 2

Reviewer 3 Report

I think the authors address my concerns. 

Congratulations

Author Response

Thank you for your comment.